# Gaps and Opportunities in the Diagnosis and Treatment of Pancreatic Cancer

**DOI:** 10.3390/cancers15235577

**Published:** 2023-11-25

**Authors:** Miłosz Caban, Ewa Małecka-Wojciesko

**Affiliations:** Department of Digestive Tract Diseases, Medical University of Lodz, 90-153 Lodz, Poland; ewa.malecka-panas@umed.lodz.pl

**Keywords:** early detection, morbidity, mortality, pancreatic cancer, pancreatic ductal adenocarcinoma, screening

## Abstract

**Simple Summary:**

Pancreatic cancer, including primarily pancreatic ductal adenocarcinoma (PDAC), represents a highly lethal disease and an increasingly common cause of cancer mortality worldwide. Diagnostics and treatment of PDAC represent a great challenge for modern medicine. PDAC is a highly malignant diagnosis, characterised by rapid local progression and distant metastasis formation. The screening of disease in general population is not cost effective but may be considered in high-risk groups. Recently, there has been great interest in developing new PDAC serum biomarkers: diagnostic, prognostic, and predictive. The most common management of early stage PDAC is radical surgery, which is rarely possible upon diagnosis. There is a strong need for further research to improve PDAC early detection and management.

**Abstract:**

Pancreatic cancer is one of the leading causes off cancer-related deaths globally. In Europe, this type of cancer has the lowest survival rate of all cancers. A majority of patients have unresectable or even metastatic disease. In addition, actual therapeutic options are not curative, and surgical treatment is associated with high post-operative morbidity and a lack of uniform translation of surgical success into long-term survival. Moreover, there is no screening for the general population which is recommended, and the overall poor prognosis in pancreatic cancer is related to late clinical detection. Therefore, early diagnosis and early treatment of pancreatic cancer are particularly critical. In this review, we summarize the most significant gaps and opportunities in the diagnosis and treatment of pancreatic cancer to emphasize need for improvement of early detection and the therapeutic efficacy of the available treatment for this cancer. Novel, inclusive, and intentional research is needed to produce improvements in pancreatic cancer in mm the world.

## 1. Introduction

Pancreatic cancer (PC) is a lethal disorder characterized by poor prognosis and response to treatment. In the past three decades, a more than 2.5-fold increase in the global annual number of PC cases has been reported. In the years 2020 and 2017, there were approximately 495,773 and 441,000 new identified PCs worldwide, respectively, in comparison to 196,000 subjects in the year 1990 [1,2]. Globally, PC has been identified as the leading cause of cancer-related deaths [3]. In 2019, PC was the fourth cause of cancer-related death in Japan [4]. It is estimated that PC will be the second-leading cause of death associated with cancer in the United States by the year 2030, surpassing prostate, breast, and colorectal cancers [5]. It results from the biology, high malignancy, late diagnosis, inefficient therapy and the lack of screening possibilities in the general population. In addition, PC is associated with an insidious clinical course and unspecific symptoms at the initial stages of the disease [6,7]. It may be incidentally detected at the advanced stage when still asymptomatic [8]. It must be emphasized that surgical resection is the only therapeutic option giving a chance for longer survival. Nevertheless, surgical resection and standard chemotherapy result in a 5-year survival rate lower than 30% [9]. In addition, only 20–30% of patients with PC are diagnosed at the resectable stage of the disease [10]. In consequence, the general 5-years survival rate of PC is significantly lower and is estimated at only about 10% [11]. Despite such unfavourable data, screening of the asymptomatic adult population is not viable or recommended with current modalities due mainly to the low incidence of the disease, the lack of simple diagnostic tests or treatment not being cost-effective About 90% of all cases with PC are sporadic, while 10% of cases develop in patients with hereditary syndromes or familial predisposition. The screening of individuals in high-risk groups is recommended, but its efficacy is not satisfactory and there are no detailed schedules elaborated [12,13]. New biomarkers of PC are constantly being searched for. Due to the fact that pancreatic ductal adenocarcinoma (PDAC) is the most common among patients with PC, we focused on data in the context of PDAC in this article.

This article discusses major gaps and opportunities in the diagnosis and treatment of PDAC. Also, the review was prepared to highlight the necessity of devoting more attention to PC that requires improvement of therapeutic efficacy and early detection.

## 2. Gaps in the Diagnosis of PDAC

### 2.1. Unspecific Symptoms

Generally, typical symptoms of PDAC, such as jaundice, weight loss, main pancreatic duct obstruction, or cholangitis associated with PDAC, appear at the advanced, non-resectable stage of the disease. At the early stage, the disease is usually asymptomatic or accompanied by nonspecific symptoms, such as epigastric or back pain, nausea, fatigue, bloating, abdominal fullness or change in stool consistency. In consequence, they are often understandably attributed to alternative, more frequent causes, causing a delay of adequate diagnosis. In turn, data show that the clinical symptoms occurring with the highest frequency at the time of diagnosis include abdominal pain, abnormal liver function tests, jaundice, new-onset diabetes, dyspepsia, nausea, vomiting, back pain and weight loss. It is worth emphasizing that clinical presentation of the disease depends on the location of the tumour within the pancreas. Biliary obstruction and the ensuing jaundice are typical for the cancer of pancreas head. Cancers of the pancreatic body are characterized by the tendency to invasion of local vascular structures including the celiac, hepatic, and superior mesenteric vessels and the portal vein. In addition, they are more likely to cause back pain. In turn, pancreatic tail cancers are often advanced at the time of diagnosis, may often grow unimpeded due to fewer anatomical barriers, and symptoms often result from metastases. It should be mentioned that cancer causing obstruction of the main pancreatic duct may induce symptoms of pancreatic enzyme insufficiency, or malabsorption of fat, contributing to post-prandial abdominal pain, flatulence, steatorrhea and deficiency of fat-soluble vitamins [14]. In a prospective cohort study, Walter et al. proved that symptoms, such as indigestion, back pain, diabetes, self-reported anxiety or depression, were associated with long total diagnostic interval (time between the occurrence of the symptom and diagnosis). It indicates that special vigilance should be maintained when these symptoms occur [15]. Primarily, proton pump inhibitor-resistant pain and back pain not associated with a disease of the musculoskeletal system may suggest pain related to PDAC. Interestingly, the chronic use of proton pump inhibitors is a risk factor for PDAC occurrence and may contribute to the development of the cancer by, among other factors, hypergastrinemia or low secretion of gastric acid promoting bacterial growth and secretion of nitrosamine stimulating pancreatic cell overgrowth [16].

### 2.2. Inadequacy of Radiological and Endoscopic Diagnosis

Radiological imaging (US, CT, MRI) and EUS play a key role in detection, staging, differential diagnosis or follow-up of PC. EUS and MRI are suggested as methods useful for screening high-risk individuals. EUS is characterized by higher sensitivity for PDAC tumour detection of less than 2 cm diameter compared to CT (95.2% vs. 42.8%) [17,18]. In turn, EUS may not distinguish malignant from benign lesions without biopsy (EUS-FNA). It results from the fact that the majority of pancreatic tumours, even benign, have a hypoechogenic appearance. For differentiating between inflammatory masses and malignancies, endoscopic ultrasound-guided fine-needle biopsy (EUS-FNB) seems to have higher diagnostic accuracy and sensitivity than EUS-FNA (93.0% and 86.6% vs. 83.6% and 69.5%). In consequence, it should be the preferred technique for diagnosis of cancer, especially in the setting of chronic pancreatitis that may simulate PDAC as pseudotumoral masses [19]. It is worth emphasizing that the choice of needle in EUS-FNB significantly affects diagnostic sensitivity. An appropriate needle should enable preservation of the cellular architecture, histological core procurement, and its use should be safe and easy. The end-cutting FNB needles are highly performing, and their use is associated with improved control at the puncture site and stability at the tip, allowing for enhanced penetration. However, their availability is limited [20]. The choice of an adequate needle determines appropriate tissue acquisition that is still impaired by the relatively high rate of false negatives dependent on inadequate tissue samples, representing remarkable gaps in the diagnostic process of PDAC. It is caused mainly by the consistent presence of fibrosis and necrosis inside the pancreatic mass, hindering proper diagnosis [21].

Different methods are characterized by various capabilities for the detection of early PDAC. A multicentre retrospective study conducted by Kanno et al. revealed several key points concerning the diagnosis and management of early-stage PDAC in a group of 200 cases at stages 0 and I, 20% of which were symptomatic. Analysis showed that the diagnostic accuracy of transabdominal ultrasound (TAUS), CT, MRI, and EUS was 67.5%, 98.0%, 86.5%, and 86.5%, respectively. In addition, a dilation and irregular stenosis of the main pancreatic duct (MPD) are main symptoms of early PDAC that may be overlooked and interpreted as not associated with malignancy. Interestingly, Kanno and co-researchers revealed also that local fatty changes of the pancreatic parenchyma occurred in stages 0 and I of PDAC with a frequency of 42% and 41.8%, respectively. Those are not the specific changes, however their detection may be associated with the possibility of PDAC and should suggest an extended diagnostics [22]. The search for new imaging techniques advancements, and their development enabling adequate identification of non-characteristic features of PDAC, including artificial intelligence, may facilitate the diagnosis of the disease. In turn, data demonstrate that contrast-enhanced endoscopic ultrasound (CEH EUS) is characterized by higher efficacy in the detection of PDAC compared to the conventional EUS: a prospective study found that the CEH EUS had a higher sensitivity (94.5% vs. 83.1%) and accuracy (84.1% vs. 78.6%) [23]. Also, the meta-analysis conducted by Yamashita and co-workers confirmed these results. This research group estimated the sensitivity and specificity of CEH EUS for PC diagnosis, at the level of 93% (95% CI = 0.91–0.95) and 80% (95% CI = 0.75–0.85), respectively [24]. It shows that new imagining technologies are necessary to improve detection of PDAC. Nevertheless, the use and operation of EUS and CEH EUS are not easy, and the accuracy of the method is strongly dependent on operator experience and qualifications [25,26,27].

### 2.3. Unsatisfactory Screening

PDAC belongs to one of the most common causes of cancer-associated death. Despite this, screening for the general population is not recommended [18,28]. Enriching the population by targeting subgroups at greater risk for PC, including PDAC, could enhance the diagnostic yield of screening. Actually, PDAC surveillance focuses on genetically predisposed subjects. It is estimated that about 10% to 15% of PDAC are due to inherited mutations. High-risk cases may be offered surveillance, depending on genetic mutation. The risk of PC for individual genetic mutations is presented in Table 1. Actually, an accepted threshold to consider PDAC surveillance is a lifetime risk of 5%.The International Cancer of the Pancreas Screening (CAPS) Consortium suggests surveillance for individuals with familial risk, from the age of 50–55 or from 10 years earlier than when the youngest relative with PC was diagnosed. High-risk of PDAC for subjects eligible for surveillance is defined in Table 2. EUS, MRI, MRCP are the recommended surveillance tests [29]. Surveillance programs in both Europe and the United States have showed that surveillance of high-risk individuals, using EUS and MRI/MRCP may be associated with the early detection of PDAC with very high rates of resectability (75–90%) [30,31]. However, it should be emphasized that the relative benefit of surveillance in families with familial pancreatic cancer is not evident. In addition, despite screening, PDAC at the advanced stage is detected too often, and the five-year survival rate is not significantly increased [30,31]. In contrast, for the general population, the screening is not recommended. It results from some facts, such as the lack of cost effectiveness, actually low incidence of PC, lack of simple biomarker or limited availability and invasiveness of EUS [32].

So far, the CA 19-9 remains the only approved biomarker for diagnosis and response assessment, but its utility is limited by low sensitivity and specificity. Despite a numerous studies searching for novel biomarkers in single and in panels of PDAC, none of them has been established for use in clinical practice. Interestingly, the combinations of markers, among others CA 19-9 and the ratio of neutrophils to lymphocytes, may turn out to be more effective and accurate compared to single biomarkers in the diagnosis and prognosis of PDAC. It was proven that the panels of biomarkers may be characterized by high levels of sensitivity, specificity and overall accuracy, and represent potential in the early diagnosis of PDAC. Nevertheless, so far, there is no recommended combination of biomarkers accepted for diagnosis and prognosis of PDAC [33,34].

In addition, differentiation between PDAC and chronic pancreatitis (CP) is also difficult with CT, MRI and MRCP. In consequence, novel biomarkers, especially measurable in easily available biological material, may represent significant diagnostic progress. Actually, there is no single biomarker clearly differentiating PDAC from CP. Nonetheless, some studies demonstrated that particles, such as soluble urokinase-type plasminogen activator receptor, osteopontin or soluble AXL receptor tyrosine kinase, may be considered for this purpose [35,36,37]. In addition, many cytokine panels together with inflammatory biomarkers have been proposed for discriminating autoimmune pancreatitis (AIP) and PDAC. Significantly higsher levels of IL-1β, IL-7, IL-13, and granulocyte colony-stimulating factor (G-CSF) were observed in patients with AIP in comparison to subjects with PDAC [38].

New-onset diabetes is actually an early symptom of PDAC often overlooked by clinicians. Also, it should be remembered that patients with PDAC may have a higher risk of diabetes. In addition to conventional imaging, annual glucose testing or assessment of HbA1C is recommended to diagnose new-onset diabetes in high-risk individuals. The screening of PDAC in new-onset diabetes was proposed, but not actually introduced [39]. It results from the very high number of new cases of diabetes annually, which are mostly not associated with PDAC. There is a strong need to find the biomarker differentiating PDAC-associated diabetes from type 2 diabetes mellitus.

## 3. Gaps in the Treatment of PDAC

### 3.1. Gaps in the Surgical Treatment of PDAC

The surgical treatment may result in numerous complications, both associated with surgery itself and anaesthesia, such as organ space/deep surgical site infection, wound dehiscence, fistulas, sepsis, stroke, coma, myocardial infarction, venous thromboembolism, pneumonia, renal failure or infection. In addition, intra-abdominal abscess, haemorrhage, fistulae of the pancreatointestinal anastomosis, delayed gastric emptying are other surgical complications, more specific for pancreatic resection. Data show that the post-operative mortality in non-specialized centres may amount to 20%, however, it is about 6% or less in specialist centres [40]. It is worth emphasizing that the most recent data indicate that the Whipple procedure is associated with high morbidity that may reach up to even 45% [41]. Comparing data, despite the development of research and surgical techniques and a decrease of mortality rate being approximately 1% in high-volume centres, there continues to be a high morbidity in these centres [42]. Moreover, the risk of complications occurring after pancreatic resection is estimated to be about 50% [43]. Interestingly, the development of postoperative complications contributes to poorer overall survival in patients who underwent curative surgery [44].

The surgical procedure, perioperative care and the surgical strategy should be carefully planned to assess the possibility of tumour resection in detail, decrease the risk of complications occurrence, as well as primarily to avoid unnecessary surgery, impossible due to the advancement of the tumour [44]. In addition, there are data indicating that use of neoadjuvant treatment may improve oncological outcomes and reduce the risk of surgical complications [43]. On the other hand, a retrospective study performed by Zhang and co-workers revealed that laparoscopic pancreaticoduodenectomy may be effective and feasible for managing selected patients with PDAC. This kind of operation is associated with decreased risk of the occurrence of postoperative pneumonia and abdominal infection in comparison to open pancreaticoduodenectomy [45]. Also, it is worth to emphasizing that proper qualification to the surgical treatment and careful choice of potential candidates have significant roles in adequately planning surgery and avoiding the impossibility of performing excision of a tumour due to non-resectability being confirmed during operation. This situation may result from difficult differentiation of necrosis, fibrosis, and oedematous tissue from malignant tumour cells, as well as the lack of reliable assessment of the extent and infiltration of cancer tissue demonstrated in preoperative imaging modalities [46].

### 3.2. Gaps in the Chemotherapy of PDAC

Over the years, new chemotherapy schemes have been developed. Chemotherapy is associated with a lot of side effects. New chemotherapeutics with low toxicity are still sought after. There are some schemes of chemotherapy for PDAC depending on the advancement of the disease: metastatic disease, resectable PDAC or locally advanced irresectable disease [47]. Barnes and co-researchers identified educational gaps according to the treatment of PDAC among oncologists in the United States. They demonstrated that 44% of oncologists did not select an evidence-based adjuvant chemotherapy regimen in patients with resectable PDAC. In turn, 57% of oncologists did not choose an evidence-based second-line chemotherapy regimen, and 35% selected a regimen containing oxaliplatin, a chemotherapeutic causing neuropathy, in a group of subjects who developed neuropathy after first-line nab-paclitaxel/gemcitabine followed by chemoradiation. In contrast, in the group with metastatic disease, only 34% of oncologists recommended a biopsy, chest imaging, and liver function tests which should be the standard of care assessments with this manifestation [48]. This study revealed significant educational gaps and discrepancies between oncologists’ recommendations and standard evidence-based guidelines. Also, Hamad et al. showed disparities in stage-specific guideline-concordant cancer-directed therapy for patients with PDAC. Adherence to stage-specific standard of care treatment occurred in only 45.3% of subjects among the entire cohort and varied by stage of disease. The use of standard therapy resulted in lowered risk of death by about 47% in comparison to subjects who did not receive standard of care treatment. The discrepancies were the most frequent in the group of patients with locally advanced stage disease, and especially with older age, women, African Americans, and patients with comorbidity burden. In contrast, treatment at a high-volume centre and higher education level led to higher likelihood of receiving stage-specific standard of care treatment [49]. Also, Lima and co-researchers presented that the patients with PDAC treated at minority-serving hospitals were less likely to obtain National Comprehensive Cancer Network-compliant care, whose the lack is related to worse overall survival and reduced long-term oncological outcomes [50].

### 3.3. Pancreatic Exocrine Insufficiency

A lot of patients with PDAC are characterized by malnutrition and weight loss, nutrient deficiencies, and associated-cachexia. It reduces quality of life and limits the effectiveness of anti-cancer treatment. Pancreatic exocrine insufficiency (PEI), resulting from inappropriate secretion of pancreatic enzymes, belongs to the significant causes of malnutrition in PC, including PDAC [51]. Systematic review and meta-analysis from 2020 presented that PEI may affect even 72% patients with advanced PC. It is worth emphasizing that pancreatic head tumours are more likely to coexist with PEI compared to the tumours of body and tail [52]. In addition, surgical treatment may lead to PEI and it is dependent on the type of surgery. Roeyen et al. conducted a prospective cohort study and showed that 64.1% of patients required pancreatic enzyme replacement therapy (PERT) after pancreaticoduodenectomy performed for oncologic indications [53]. By definition, it is important to remember that all patients after total pancreatectomy will have PEI and require PERT. Generally, pancreaticoduodenectomy is related to increased risk of PEI compared to the distal pancreatectomy [54]. Preoperative diameter of the main pancreatic duct greater than 3 mm, hard pancreatic texture, and the use of adjuvant chemotherapy were identified as risk factors for the development of PEI in PDAC [55]. Noteworthy is the fact that the detection of PEI in PDAC is difficult, and the problem is often overlooked.

Generally, the diagnosis of PEI in PDAC is based on faecal elastase-, dietary concentration, and further dietary consultation and the use of PERT is being introduced. Comprehensive nutritional assessment, intake of smaller and more frequent meals, replacement of any deficiencies in fat-soluble vitamins, vitamin B12, iron, and lipoproteins should follow. PERT should be used in adequate dosing: recommended starting dose is approximately 40,000–50,000 units of lipase with meals and 25,000 units of lipase with snacks. Alternative doses are 500–2500 units of lipase/kg/meal, half for snacks, to a maximum of 10,000 units of lipase per kg per day [51]. Nevertheless, even adequate use of PERT cannot lead to the complete alleviation of clinical symptoms of PEI. This is due to the numerous interactions between pancreatic maldigestion, intestinal inflammation or intestinal microbiota [56].

In a retrospective study, Landers and co-researchers revealed that 72% of patients with metastatic PDAC had symptoms of PEI. However, only 21% of them supplemented PERT. This is probably due to the lack of awareness of PEI in PDAC, as well as the benefits of PERT to long-term outcomes, and insufficient screening [57]. It is worth to emphasizing that there are conflicting data about the improvement of nutritional status, quality of life and overall survival in patients with PDAC receiving PERT [51,58,59,60]. The contradictory evidences is limited by the small study groups, heterogeneity of the PDAC patients population (e.g., tumour localization, the presence of metastatic variant of the disease, tumour resectability), and availability of specialized dietary support for the patients in individual health centre. Generally, PERT in PDAC seems to be useful and improves patients’ survival. Nevertheless, the use of PERT in PDAC is too often overlooked by clinicians.

## 4. Opportunities in the Diagnosis of PC

### 4.1. Acute Pancreatitis

Acute pancreatitis (AP) is an inflammatory disorder characterized by high mortality. In addition, the disease is one of the leading gastrointestinal causes of hospitalisation [61]. AP may be caused by the underlying PDAC.

It is well established that recurrent AP may lead to chronic pancreatitis which is a well-known PDAC risk factor. Over the past decade, numerous studies evaluated AP as a risk factor for PC. Munigala and co-researchers conducted a retrospective study using a nationwide Veterans Administration database spanning 1999–2015. They revealed that there was increased risk of PDAC 3–10 years after AP irrespective of the aetiology [62]. In addition, in a nationwide, population-based, matched cohort study, Kirkegård et al. demonstrated that subjects with AP had an elevated risk of PDAC compared to the age- and sex-matched general population [63]. Moreover, there are two meta-analyses showing the increased risk of PDAC after AP. However, AP is unlikely to be a causative factor for PDAC [64,65]. Taking into account that the risk of developing PDAC after AP decreases with time (a strong relationship between AP and PDAC occurs in the first two years after AP), it seems that AP has no causal correlation with PDAC. Therefore, AP may be sometimes the first symptom of the cancer causing obstruction of the pancreatic ducts and accompanying inflammation [66]. It is estimated that approximately 1% of AP is caused by PDAC that is usually diagnosed within a few months after the diagnosis of AP [67]. Most cases of AP caused by PDAC have a mild clinical course, which may result in limited diagnostics towards a potential malignancy. Mild AP turned out to be a predictor of PDAC, while severe AP was not associated with PDAC [67,68]. Therefore, in patients with mild AP without an identifiable cause, a possible underlying PDAC may be suspected. Furthermore, dilation of the pancreatic duct, anaemia, age over 50, newly diagnosed diabetes mellitus, evidence of CP features in imaging tests or weight loss may be factors indicating on PDAC as a cause of AP with undetermined aetiology [68,69,70]. Interestingly, cancers identified in patients with AP tend to be in the early stages, are smaller in size and more often resectable, leading to lower mortality compared to patients without a history of AP. This is caused by the earlier onset of clinical symptoms due to the painful AP episode [67,71,72,73]. Therefore, in all patients with AP without an identifiable cause, serum measurement of CA 19-9 may be considered. Teng and co-workers proved that significantly increased level of CA 19-9 in patients with AP predicts the presence of PDAC [74]. Nonetheless, the normal levels of this marker do not exclude the cancer disease.

On the other hand, there is also evidence showing AP preceding PDAC resection, especially moderate and severe, as a condition that may reduce the patient survival. It results from severe peripancreatic inflammation causing surgical resection to be difficult with higher rate of postoperative complications [75,76].

### 4.2. Diabetes Mellitus

Data indicate that there is bidirectional association between diabetes mellitus (DM) and PDAC. In addition, it is proved that DM may be independent risk factor, as well as a complication of PDAC. It is estimated that approximately 85% of the patients with PDAC have impaired glucose tolerance or even diabetes. Type 2 DM occurring in the elderly usually over 65 years, new onset diabetes (less than 30 days) in elderly patients, heavy smoker, low BMI, history of CP, weight loss associated with DM onset are alarming signs and indicators that may suggest PDAC [77,78]. In turn, about 0.8–1% of individuals aged over 50 years with new-onset diabetes generally have diabetes secondary to PDAC [13].

Patients with type 2 DM lasting over 5 years have a 1–1.5-fold elevated risk of PDAC compared to the general population. In contrast, the relative risk of PDAC occurring in subjects with type 2 DM in less than 1 year increases to even 5.4-fold [79]. Also, another meta-analysis demonstrated a 6.69 times increased risk of PDAC in new onset diabetes, particularly within first year of diagnosis [80]. Interestingly, new onset DM associated with PDAC is diagnosed less than 24–36 months before diagnosis of PDAC, and it’s course is mitigated by surgical resection of the tumour causing primarily a reduction of fasting blood glucose [81]. Dankner and co-researchers revealed a high risk of PDAC development in both women and men (HR of 15.24 and 13.88 respectively) in the first year after the DM diagnosis in a large cohort of 2.3 million Israelis [82]. Aggarwal et al. observed a very high prevalence of DM (68%) in patients with PDAC compared to age matched other cancers subjects or non-cancer controls [83]. It is worth emphasizing that the occurrence of dysglycaemia in PDAC was detected more frequently with the standard oral glucose tolerance test (OGTT) than fasting glucose levels (78% vs. 45%) [78,83,84]. On the other hand, a significant number of new onset DM in patients with PDAC regressed after tumour resection [85]. Those data may indicate that new onset DM may represent an early biomarker for PDAC. The detailed mechanisms underlying this coincidence remain unclear. However, it is considered that PDAC may lead to DM through releasing cancer cell mediators stimulating hyperglycaemia or β-cell dysfunction and enhancing insulin resistance, such as adrenomedullin, islet amyloid polypeptide. DM is associated with β-cell dysfunction mediated by inflammation, β-cell loss occurring as a consequence of oxidative stress and aberrant activation of the inflammation- and cancer-associated pathways, inducing cell apoptosis [81]. Moreover, elevated levels of glycated haemoglobin (HbA1c) being a mark of deterioration of the existing glycaemic control may be associated with the development of PDAC. Lu and co-workers proved that the level of HbA1c has been rising from the diagnosis of DM to detecting PDAC definitely more severely than in patients without diagnosis of PDAC after detecting DM. It shows that rising HbA1c may be an independent risk factor for PDAC [84].

The routine use of radiological imaging to detect PDAC after the DM diagnosis is not cost-efficient. Imaging in this group of patients does not fulfil the criteria of screening, and the incidence of DM in the general population is too high and in the vast majority of cases, there is no PDAC. Nevertheless, imaging diagnosis, including MRI/CT of the pancreas, can be dedicated to middle age and elderly patients (≥50 years) developing DM, as well as in those with rapidly altered glycaemic indices, body weight loss in the preceding months prior to DM onset, absence of typical signs of the metabolic syndrome or of family history of PDAC [86,87]. Data derived from the next next studies revealed clinical prediction models determining high-risk individuals among those with newly diagnosed DM who might benefit from PDAC screening. Sharma et al. developed a model, including age at diabetes diagnosis and change in weight and blood glucose, and stratified patients into three risk groups. The sensitivity and specificity of the cut-off for the high-risk group was 80% [88]. In contrast, Boursi et al. validated two models. The first of them takes into account inter alia age, BMI, change in BMI, smoking, use of anti-diabetic medications, serum levels of HbA1C. Its sensitivity and specificity were 45% and 94%, respectively [89]. The second model comprised age, BMI, proton pump inhibitor use, total cholesterol, low-density lipoprotein, alanine aminotransferase and alkaline phosphatase and was dedicated to patients with impaired fasting glucose. The model achieved good discrimination (area under the curve 0.71 (95% CI, 0.67–0.75) [90]. Nevertheless, DM may be helpful and suggestive of diagnosis of PDAC. However, further studies are necessary to identify new markers distinguishing PDAC-associated diabetes from typical type 2 DM.

### 4.3. New Techniques of Imaging

Konno et al. proved that intrapancreatic late enhancement of pre-diagnostic contrast-enhanced CT is relevant in the early detection of PDAC. Additionally, it occurred significantly more frequent in PDAC compared to other indirect findings of cancer, such as MPD abnormalities or focal pancreatic parenchymal atrophy. Consequently, intrapancreatic late enhancement without other indirect findings can facilitate detection of PDAC, mainly located in the head of pancreas [91]. Real-time elastography (RTE), a novel method for the assessment of tissue elasticity using ultrasound, is used in the diagnosis of various type of tumours to differentiate malignant from benign lesions. There are two semi-quantitative elastography methods: mean strain histograms (SH) and strain ratio (SR). The mean SH value corresponds to the hardness of the lesion depicted by the colour on the scale from hardest to softest. The novel diagnostic option of PDAC is shear-wave elastography (SWE) that enables quantification of mechanical and elastic tissue properties: tissue of PDAC is stiffer than the healthy parenchyma [92]. Another valuable EUS-related technique for the diagnosis of PDAC is contrast-enhanced EUS (CE-EUS). The use of contrast enhancers could provide additional information about the vascularization of the tissue and facilitate the diagnosis of malignant lesions in the pancreas. CE-EUS is able to differentiate PDAC from neuroendocrine tumours (NET), as well as primary pancreatic tumours from metastases. Generally, NET is characterized by hypervascularization and a low microvessel architecture, whereas PDAC has iso-enhancement or hypo-enhancement, arterial irregularity and absent venous vasculature within a mass. In turn, metastases are mostly hypervascular, however they may be both hypernehanced and hypoenhanced depending on type and localization of the primary tumour. In addition, this method is characterized by higher accuracy for the diagnosis of small PDAC compared to contrast-enhanced multidetector CT [92].

The combination of RTE and EUS-FNA had higher diagnostic accuracy, sensitivity, and specificity compared to EUS-FNA alone [93]. In addition, it was evidenced that pathological assay combined with analysis of KRAS-mutation using allelic discrimination demonstrated higher accuracy (88% vs. 73%) and sensitivity (93% vs. 85%) for PDAC diagnosis compared to cytopathology alone [94]. Also, novel optical system-spatial-domain low-coherence quantitative phase microscopy (SL-QPM) enhanced the accuracy of EUS-FNA cytological diagnosis and elevated the sensitivity of cytology for identifying PC from 72% to 94% [95].

### 4.4. Artificial Intelligence

Artificial intelligence (AI) is a complex integration of a computational machine system, whose aim is the replication of human intelligence to carry out given tasks and improve themselves in accordance with the information they collect. Actually, clinical practice is revolutionized by three main branches of AI, such as machine learning (ML) (including deep learning (DL)), artificial neural networks (ANN) and expert systems (ES). The use of AI may facilitate converting data into functionality perception, increasing diagnostic efficacy and predicting real-time prognosis [96,97]. Most studies concerning AI in the context of PDAC have focused on classifying cancer and healthy pancreas pictures using CT scans or differentiating PDAC from radiological mimickers in AI-assisted EUS. The AI may evaluate thousands of photos on a pixel-by-pixel basis. In addition, it is not vulnerable to human error. The rapid diagnosis is another advantage of AI. It takes less than a minute from the original CT picture being entered to diagnosis. The most recent studies used AI to PDAC diagnosis in CT, however its sensitivity and accuracy were not clearly determined [98,99,100].

In addition, AI could be applied in the prognosis of PDAC. Research performed by the Alliance of Pancreatic Cancer Consortium Imaging Working Group aimed to extract the pre- and post-diagnosis CT, MRI, and transabdominal ultrasound images from subjects who are eventually diagnosed with PDAC. The images will be used to create repository which may be shared and examined later, and it may contribute to constructing models of AI diagnosing PDAC at the early stages and calculating the advent of the cancer [101]. The analysis of real-world clinical records using ML approaches is another possible platform for introducing ML to the need for early detection of PDAC [102].

In the context of EUS, a lot of studies have been conducted to determine and compare the diagnostic accuracy of non-AI- and AI-enhanced models of EUS for PDAC. For example, a retrospective study performed by Kuwahara and co-researchers revealed that the sensitivity, specificity, and accuracy of the DL algorithm using EUS images as input data for detection of IPMN malignancy were approximately 95.7%, 92.6%, and 94.0%, respectively. It should be emphasized that the accuracy of this AI algorithm for identifying malignant probability was higher than human diagnosis and determination based on the presence of mural nodules in the tumour [103]. In contrast, another retrospective study assessed the utility and ability of support vector machine (SVM) AI-assisted EUS to differentiate PC from normal tissue. All patients underwent EUS-guided FNA and pathologic analysis. The researchers proved that the SVM model was characterized by the accuracy, sensitivity and specificity of 98%, 94.3%, and 99.5%, respectively, suggesting its strong potential for PC screening [104]. Furthermore, a systematic review from 2022 disclosed the general accuracy, sensitivity, and specificity in the ranges of 80–97.5%, 83–100%, and 50–99%, respectively. It indicates the high potential of AI-assisted EUS models as diagnostic tools for the detection of PDAC [105].

It is worth to emphasizing that differentiating PDAC from CP or AIP is the diagnostic challenge. It results from the fact that CP and AIP have characteristic features in sonographic and cross-sectional radiological imaging mimicking PDAC [96]. Nevertheless, there are data suggesting the use of AI-assisted EUS models to improve the sensitivity and specificity of EUS in differentiating CP and AIP from PDAC among other by using ANN, convolutional neural networks (CNN) or extended neural network (ENN) [106,107,108]. In spite of many advantages, the use of AI has also some limitations, including statistical management, small size of the input data, unsatisfactory quality of the input data, inefficient data handling, inadequate standardization and validation or ethical and legal aspects [97].

### 4.5. Circulome in PDAC

The “circulome” is described as a collection of circulating cancer cells or various particles, molecules, metabolites, such as growth factors, proteins, fatty acids, characteristic off the disease. For example, the prostate-specific antigen (PSA) is well-established marker for prostate cancer used for screening. In turn, CA 19-9 may be used in the PDAC diagnostics, however it is not useful in the screening of PDAC.

In the blood of patients with PDAC, cell-free nucleic acids (cfNAs), such as DNA, mRNA, and noncoding RNA may also be used to monitor PDAC. Circulating tumour DNA (ctDNA) derives from primary tumour or metastases and occurs in the serum as a consequence of apoptosis, lysis or necrosis of cancer cells. Therefore, ctDNA may be valuable biomarker of cancer and applied for detection of the disease. In addition, profiling ctDNA, including quality assessment of cancer genomes such as mutations, amplifications, deletions and translocations, may facilitate the identification of genetic alterations for patient stratification, and even therapeutic response prediction [109]. Moreover, the amount of ctDNA may correlate with a higher number of liver metastases, the presence of lung and/or peritoneal metastases, tumour burden and higher CA 19-9 levels [110]. In addition, ctDNA may be useful for the identification of cancer relapse: the detection of ctDNA is up to 10 months earlier than the disclosure of circulating tumours cells [111,112]. In the context of PC, a lot of studies were performed to assess the utility of ctDNA as a biomarker in PDAC. Most of them utilized K-RAS mutations to target ctDNA. Nevertheless, the sensitivity and specificity were not satisfactory and ranged between 27–81% and 33–100%, respectively [113,114,115,116]. On the other hand, the presence K-RAS mutation in serum cell-free ctDNA is unfavourable and associated with significantly shorter survival than that of patients without mutations [117]. Nevertheless, the association between the presence of K-RAS mutation in ctDNA and poor prognosis of patients with PDAC was confirmed in some studies [113,118]. Besides gene mutation, DNA methylation is an emerged diagnostics in liquid biopsy. Shen and co-researchers demonstrated that the methylation patterns in ctDNA may detect PDAC, even tumours at early stage [119]. The meta-analysis conducted by Guven et al. revealed that positive preoperative or postoperative ctDNA was correlated with lower relapse-free survival/progression-free survival (HR: 2.27, 95% CI: 1.59–3.24, *p* < 0.001) and overall survival (HR: 2.04, 95% CI: 1.29–3.21, *p* = 0.002) in localized PC. Similarly, positive baseline ctDNA was related to lower relapse-free survival/progression-free survival (HR: 2.61, 95% CI: 1.94–3.51, *p* < 0.001) and overall survival (HR: 2.41, 95% CI: 1.74–3.34, *p* < 0.001) in advanced PDAC [120]. Therefore ctDNA analyses may serve a leading role in streamlining candidate biomarkers of PDAC and could be a promising tool to individualize treatment planning and improve outcomes [121]. Unfortunately, this method is actually difficult to access.

In turn, miRNAs are small non-coding RNA molecules of about 18–22 nucleotides regulating gene expression by mRNA degradation or translatory inhibition. In addition, they play significant roles in oncogenesis and tumour metastasis, and the expression pattern of miRNA is unique to cancer type and may be a rich source of data about tumour pathogenesis. Moreover, the most recent data demonstrate that miRNAs may be used as biomarkers or even prognostic factors for many cancers, including pancreatic cancer. It is worth emphasizing that stability in serum and a convenient method of miRNA detection are the main advantages of using miRNA in the context of PDAC [122]. Eun Jo Lee et al. using pancreatic cancer, paired benign pancreatic tissue, a normal pancreas, and pancreatitis tissues along with nine cell lines and revealed miRNA expression profiles in PDAC. They showed that miRNA-345, miRNA-142-P, miRNA-139, miRNA-424, miRNA-100, miRNA-301, miRNA-212, miRNA-125b-1 were aberrantly expressed. The expression of first three was downregulated, whereas the expression of relevant molecules was upregulated in cancer samples compared to normal pancreatic samples [123]. Other data showed the overexpression of miRNA-196a, miRNA-190, miRNA-186, miRNA-221, miRNA-222, miRNA-200b, miRNA-15b, and miRNA-95, miRNA-21, miRNA-26b, miRNA-194, miRNA-200b, miRNA-200c, miRNA-320, miRNA-374 and miRNA-429 in PC tissue or PDAC cells [124]. In contrast, miRNA-96, miRNA-217, miRNA-141, miRNA-20a, and miRNA-29c were characterized by downregulated expression in PDAC samples compared to normal pancreatic tissue [125]. Furthermore, increased expression of miRNA-155 and -21 was also revealed in IPMN, one of the precursor lesions for PDAC [126]. Above data suggest a potential significance of using miRNAs as biomarkers PDAC. Also, there is evidence presenting miRNA as prognostic marker of PDAC. Also, the assessment of miRNA expression may facilitate determination of chemoresistance and responsiveness to different therapeutic options. For example, increased levels of miRNA-200c was associated with higher post-surgery survival rate [127]. In contrast, lowered expression of miRNA-200b, miRNA-200c, let-7b, let-7c, let-7d and let-7e occurred in gemcitabine-resistant cells of PDAC [128]. Described data indicate that miRNA in serum, pancreatic juice, pancreatic tissue or fluid of pancreatic cysts may represent the predictive and prognostic markers for PDAC. However, further studies are needed to establish their utility in PDAC diagnosis and prognosis.

Another component of circulome is circulating tumour cells (CTC) being cells with low abundance in the blood stream. They are detached from the primary tumour and are the main source of metastases [129,130]. Generally, CTC number in PDAC patients was an independent indicator for worse progression-free survival [131]. Unlike ctDNA, CTCs may provide a lot of information, not only genetic variation, but also the expression of genes and cytoplasm proteins, of cellular contents which are preserved and garnered by the cell membrane. In addition, it is possible to conduct ex vivo CTC culture, enabling determination of personalized therapeutic response prediction and drug screening [132,133]. It was evidenced that the CTC count in PDAC correlated with staging, prognosis, tumour resectability, as well as contributing to an adequate choice of oncological therapy. CTC is an unique opportunity to examine an aggressive population of an individual patient’s cancer cells that have achieved the metastatic step of intravasation and may determine response to the treatment [134]. The main limitation of the use of circulating pancreatic epithelial cells in the blood stream is their low amount in blood at PDAC early stages. However, technologies to extract and process large volumes of blood, such as leukapheresis, have being developed [135]. Above findings indicate that cancer cells may be detected in the circulation of patients before the development advanced PDAC, and it might be used in risk assessment and the diagnostic process. Nonetheless, large-scale prospective clinical trials with circulating pancreatic epithelial cells/CTC-guided management are still necessary to assess the clinical potential of these cells.

Despite many advantages, the assessment of circulome is difficult to assess and requires extensive laboratory facilities. The assessment of mi-RNA seems to be the most promising among other components of the circulome. However, it’s actual availability is still extremely limited. Meanwhile, simple tests such as serum biomarkers, would be desirable, currently unavailable due to the complex genetic background of PDAC.

## 5. Opportunities in the Treatment of PDAC

Currently, there are new opportunities in the therapy of PDAC, both in radical and palliative treatment. In the context radical therapies, minimally invasive pancreaticoduodenectomy seems to be the procedure with lower risk of complications, therefore improving the prognosis of patients with PDAC [136]. In addition, mesenteric vascular resections with reconstructions are also possible during laparoscopic pancreaticoduodenectomy, which is challenging and requires advanced skills of both resection and reconstruction compared to the open pancreaticoduodenectomy [136,137]. There are few methods of minimal invasive pancreaticoduodenectomy, including total laparoscopic pancreaticoduodenectomy with both completed laparoscopic resection and digestive reconstruction, hand-assisted laparoscopic pancreaticoduodenectomy, total robotic pancreaticoduodenectomy with both resection and digestive reconstruction using da Vinci surgical system and robotic-assisted pancreaticoduodenectomy with laparoscopic dissection and reconstruction completed by da Vinci surgical system [138]. Randomized clinical trial from 2018 showed that the length of stay and post-operative complication burden were reduced after laparoscopic surgery compared to the open pancreaticoduodenectomy [139]. Interestingly, data from two meta-analyses, did not reveal significant differences in rates off surgical mortality and pancreatic fistulas or overall survival and 5-year survival after laparoscopic and open pancreaticoduodenectomy [140,141,142,143]. However, R0 resections and harvested lymph nodes were significantly higher in the group of laparoscopic pancreaticoduodenectomy [142,143]. In addition, robotic pancreaticoduodenectomy was accompanied with decreased blood loss and rate of severe complications as opposed to open surgery, as well as reduced wound infections and earlier activity after surgery. However, robotic pancreaticoduodenectomy was associated with longer operation time, and the number of experienced centres and surgeons is limited [144,145].

In the context of palliative care, there are opportunities in surgical, endoscopic and oncological treatment. Biliary obstruction occurring in non-resectable PDAC requires drainage with endoscopic retrograde cholangiopancreatography (ERCP) with stenting, or percutaneous transhepatic cholangiodrainage (PTCD). In patients with gastroduodenal obstruction, endoscopic therapy should be considered. However, the efficacy is limited due to frequent stent dislocation and recurrent obstruction. Nevertheless, in a lot of cases with PDAC, the failure of ERCP is observed. The most recent studies show high feasibility and effectiveness of EUS-guided biliary drainage (EUS-BD) for the palliation of jaundice resulting from distal malignant biliary obstruction (MBO). Therefore, after lack of success of ERCP, the EUS-BD should be considered as the next options for the treatment of MBO in PDAC. EUS-BD may be achieved through different approaches, depending on the experience and preference of the endoscopist, the availability of specific devices, the localization of the biliary obstruction and the accessibility of the papilla, such as EUS-guided antegrade stenting, EUS-guided choledochoduodenostomy (EUS-CDS) or choledocho-gastrostomy, EUS-guided hepato-gastrostomy (EUS-HGS), Rendez-vous (RV) technique, EUS-guided cholecysto-gastrostomy (as a last resort). Initially, luminal biliary stents dedicated to EUS intervention were used with predictable shortcomings in technical success, outcome and adverse effects. However, the development of new metal stents contributed to significant improvement of the EUS-BD results. Lumen-apposing metal stents (LAMS), new fully covered “dumbbell”-shaped short metal stents specifically constructed for transluminal drainage, have been created available for EUS-guided procedures, primarily intended for the management of fluid collections. In addition, electrocautery-enhanced LAMS (EC-LAMS), newer modification of LAMS, simplification of the classic multi-step procedure of EUS-BD allow direct access to the target lumen. The safety and effectiveness of EUS-BD using LAMS and EC-LAMS were confirmed, and this method seems to be one of the most often performed techniques for EUS-BD. However, adverse events, such as duodenal perforation, bleeding, transient cholangitis, stent obstruction, kinking, spontaneous dislodgement, food impaction, may occur. Further studies are needed to determine newer technical innovation in the design and delivery system, improving LAMS performance by increase the duration of patency and reduction of adverse effects during deployment [146,147,148].

After failure of endoscopic methods and for the remaining cases, biliary, gastroenteric or double bypass may be introduced, improving the quality of the patient’s life. However, the most recent data showed that such proceedings are associated with higher mortality and morbidity compared to explorative surgery only [149]. That is why these methods should be individually implemented with due caution. Irreversible electroporation or Nanoknife ablation being non-thermal ablative techniques are valuable therapeutic methods and associated with induction of high voltage, short pulse electrical fields, leading to local cellular apoptosis [150]. These methods may destroy tumours tissue without compromising adjacent vessels or bile ducts. In consequence, they could be applied as a potential palliative option in unresectable, locally advanced PDAC [151,152]. Martin and co-researchers conducted a pilot study enrolling patients with unresectable locally advanced PDAC and performed either resection of the primary tumour with IRE (NanoKnife^®^) margin accentuation (n = 8) or IRE alone (n = 19). Their 3 month-observation was associated with complete tumour destruction in all subjects without local recurrence [153]. There is no evidence presenting detailed data about the impact of NanoKnife^®^ on the survival of patients with PDAC. However, clinical trials assessing the safety and efficacy of the NanoKnife^®^ system for the ablation of stage 3 PDAC are in progress (NCT03899636 and NCT03899649).

Palliative chemotherapy is dedicated for unresectable and metastatic PDAC patients to improve overall survival. A lot of schemes of therapies are still assessed. Multidrug schemes have advantage over schemes with one chemotherapeutic. Phase III clinical trials demonstrated that FOLFIRINOX or nab-paclitaxel with gemcitabine improved median overall survival compared to gemcitabine alone [154,155]. Also, research on adjuvant chemotherapy after surgery in PDAC is still in progress. Improvement of overall survival from single agent-adjuvant therapy using 5-FU or gemcitabine been proven [156,157,158]. The ESPAC-4 study presented longer overall survival after the use of gemcitabine with capecitabine as adjuvant chemotherapy in comparison to gemcitabine monotherapy (28 vs. 25.5 months) [158]. In turn, PRODIGE 24, a large multinational phase III trial, revealed the superiority of a modified FOLFIRINOX regimen (reduced irinotecan and no bolus 5-FU) over gemcitabine in the adjuvant setting, improving overall survival (54.4 months vs. 35 months). Nonetheless, more toxicity was also observed [159]. Generally, patients in good general condition are administered more aggressive treatment strategies which contribute to longer overall survival, such as the FOLFIRINOX programme or gemcitabine in combination with nab-paclitaxel (albumin-bound paclitaxel). This formulation of paclitaxel increases the drug concentration in PDAC cells by 30%. There are also new available drugs. Olaparib, PARP inhibitor, was accepted by FDA in 2019 for metastatic PDAC treatment in patients with BRCA mutation. In turn, pembrolizumab, programmed cell death protein 1-targeted antibody, may be used for PDAC with microsatellite instability (MSI) as a second line-chemotherapy after detecting high MSI or mismatched repair deficiency. For the PDAC-patients with neurotrophic tropomyosin receptor kinase (NTRK) gene fusion mutation, other targeted drugs, such as larotrectinib and entrectinib (NTRK inhibitors) may be used as second-line therapy based on phase I and case series experience [160,161,162].

Over the last decade, the development and progression of studies aimed to reveal significant pathways, genes or immunological processes engaged in PDAC contributed to identification of pharmacological targets. Data show that approximately 25% of the patients with PDAC have somatic/germ line mutations in DNA damage repair (DDR) pathway (BRCA 1,2, PALB2) homologous recombination repair (HRR) genes (ATM, BAP1, BARD1, BLM, BRIP1, CHEK2, FAM175A, FANCA, FANCC, NBN, RAD50, RAD51, RAD51C, and RTEL1). The detection of DDR pathway deficiency in PDAC may be key, because these patients respond to platinum-based therapy, and they may benefit from the use of PARP inhibitors, such as olaparib [163]. In contrast, mutations of K-RAS gen, especially codon 12, are detected in about 95% patients with PDAC [164]. K-RAS therapeutics are actually subjects of numerous research. For example, agents, such as sotorasib and adagrasib, K-RAS G12C inhibitors, revealed clinically meaningful benefit and safety [163,165].

PDAC is considered immunologically “cold”, and the past studies did not reveal any response to immunotherapy. Nevertheless, studies using immunological mechanisms in the context of PDAC are still in progress. There is some evidence aimed at making tumours immunologically active by stimulation innate lymphocytes or inducing an inflammatory reaction in the tumour microenvironment. Leidner and co-researchers proved the utility of neoantigen T-cell receptor gene therapy in progressive metastatic PDAC. The infusion of genetically engineered autologous T cells led to the regression of visceral metastases [166]. Selvanesan and co-workers administered non-disease-causing Listeria bacteria carrying a gene encoding for tetanus toxin into mice model of PDAC. The tetanus toxin-induced memory T cells stimulated cytotoxic T cells in tumour microenvironment, leading to a reduction of tumour size and then low dose gemcitabine was administered [167]. There is a clinical trial (NCT04999969) assessing the role of AZD 0171 (anti LIF1 monoclonal antibody), durvalumab (PDL-1 inhibitor) in combination with gemcitabine and nabpaclitaxel in an open label phase II study. In turn, antibodies, such as ipilimumab and nivolumab, may prove to be efficient in patients with PDAC concomitant with harbouring germ line HRD genetic mutations, resistant to platinum-based chemotherapy and PARP inhibitors [168]. A lot of novel drugs dedicated to metastatic PDAC targeting metabolism, tumour stroma, oncogenic pathways and immunosuppression are in clinical development [169]. Nevertheless, the development of next-generation sequencing testing, molecular experiments and genetic assays in the context of PDAC are necessary for the development of targeted and personalized therapy in PDAC.

Neoadjuvant radiotherapy should be also developed. The PREOPANC-1 study from 2018 presented evidence of the utility of neoadjuvant chemoradiation in border-line resectable patients. The 246 patients with borderline resectable PDAC into either neoadjuvant gemcitabine with 36 Gy radiation administered between two cycles of gemcitabine or upfront surgery. It is worth emphasizing that both groups received adjuvant chemotherapy. A smaller number of patients in the chemoradiation group underwent surgical resection (62% vs. 72%). However, these subjects achieved significantly better R0 resection rate (65% vs. 31%), disease-free survival (median 11.2 vs. 7.9 months) and overall survival (median 17.1 vs. 13.3 months) [170]. Therefore, neoadjuvant chemoradiation may be an opportunity for some patients with borderline resectable PDAC. In contrast, CivaSheet, an FDA-cleared implantable sheet with a matrix of unidirectional planar low-dose-rate (LDR) Palladium-103 (Pd-103) sources, could be considered and used in the treatment of borderline resectable PDAC. Achievement of negative surgical margins remains technically challenging after pancreatectomy, and positive margins lead to increased local recurrences and worse overall survival. The CivaSheet may easily be implanted at the time of surgical treatment [171]. On the other hand, ablation treatments may be considered in the management of unresectable locally advanced PDAC without metastases. Nevertheless, expertise is required to use this method and to avoid serious or fatal complications [172]. Interestingly, irreversible electroporation has an advantage over radiofrequency ablation after induction chemotherapy for treating locally advanced PC. It is worth emphasizing that the comparable efficacy of these two therapies was presented for tumours larger than 4 cm [173].

The most recent data present photodynamic therapy as a valuable therapeutic method supporting adjuvant therapy of PDAC. It may be implemented in the treatment of a multifocal disease with the least amount of health tissue damage. The use of photodynamic therapy leads to cancer cell apoptosis, the damage of pathological vessels and even enhancement of anti-cancer immune response. However, the desired effect may not be achieved due to resistance mechanisms of PDAC cells or limitations of current photosensitizers, such as restricted penetration depth, poor targeted therapy and inappropriate generation of reactive oxygen species. Photodynamic therapy may be an adjuvant therapeutic option after surgery and combined with chemotherapy to clear residual lesions both locally and systematically. Also, the combination of photodynamic therapy with chemotherapeutics may effectively decrease their doses, resulting in reduced systemic toxicity. Moreover, photodynamic therapy may be applied as supportive therapy in metastatic PDAC and achieve satisfactory outcomes, leading to reduction of patients’ suffering. Despite described advantages of the use of photodynamic therapy in PDAC, further clinical trials are necessary to confirm its utility. It seems that the development of novel hypotoxic photosensitizers may have a key role for improvement of the application of photodynamic therapy in multimodality treatment of PDAC [174,175].

On the other hand, the drug resistance is a key factor limiting therapy effectiveness and remains a significant clinical oncology challenge. It seems that the use of nanotechnology may overcome this restriction. Data revealed that the application of nanomedicine in PDAC was related to beneficial effects, such as bypass drug efflux pumps, sensitization to radiotherapy or counteracting immune suppression, which collectively mitigated various mechanisms of tumour resistance. Mainly, in the context of PDAC, nanomedicine is tested using nanoparticles being drug delivery systems. They improve pharmacokinetics of drugs, protect against degradation and regulate controlled release, contributing to precision targeting of drugs to the tumour. So far, two clinically approved nanomedicines, Abraxane (paclitaxel) and Onivyde (irinotecan), are used in the treatment of PDAC, with numerous others demonstrating promise as agents at various stages of development. Therefore, novel bioformulations of drugs may occur important for improvement of the PDAC therapy and change the outlook for PDAC patients. Nevertheless, further clinical trials are needed and they should include co-load multiple cargoes within a single nanoparticle platform, allowing for the aggressive targeting of multiple aspects of PDAC pathophysiology [176].

Noteworthy is microbiome that plays a key role in the development and progression of many cancers. Data showed the association between oral and tumour microbiome, and PDAC. Changes in abundances of some bacteria taxa were observed. For example, McKinley and co-researchers presented translocation of oral microbiota into the PDAC tumour microenvironment. *Veillonella atypica* was the most prevalent and abundant taxon found within both saliva and tumour tissue samples. In addition, it was cultured from patient saliva, sequenced and annotated, identifying genes potentially contributing to tumorigenesis. High sequence similarity was revealed between sequences recovered from patient matched saliva and tumour tissue, indicating that the taxa identified in PDAC tumours may derive from the mouth [177]. In contrast, Kabwe et al. proved that *Porphyromonas* can promote the growth of PC. There is evidence that bacteriophages may access and treat multiantibiotic resistant pancreatic infections. Therefore, the use of bacteriophages may modulate pancreatic microbiome, reducing the development of PDAC [178]. Interestingly, the bacteria most associated with PC are *Gammaproteobacteria*, including *Enterobacter and Pseudomonas*, that may produce cytidine deaminase and stimulate the metabolism of gemcitabine into its inactive form, 2′,2′-difluorodeoxyuridine, contributing to the degradation and resistance of gemcitabine. The combination of gemcitabine and antibiotics is more efficient than gemcitabine alone [179,180]. However, the use of penicillin is closely associated with the occurrence of PDAC, and the association increased with the number of antibiotic courses. This dependence was confirmed by determining adjusted odds-ratios and 95% confidence intervals using conditional logistic regression in a nested case-control study using a large population-based electronic medical record database [181]. In turn, the application of probiotics during chemotherapy of PDAC may enhance the efficacy of chemotherapeutics. It was evidenced that probiotics combined with chemotherapy are able to increase the DNA damage of PDAC cells, inhibit the cancer cells cycle or even stimulate cell apoptosis. In addition, probiotics may restore the number of platelets damaged by gemcitabine [180,182]. On the other hand, probiotics containing *Lactobacillus* and *Bifidobacterium* have a protective effect on radiotherapy-induced intestinal toxicity and may significantly decrease the incidence of severe diarrhoea [183,184]. Interestingly, faecal microbiome transplantation is still tested in the context of PDAC [180]. Nevertheless, there is a need for intensification of studies assessing the role of microbiota, microbiome, antibiotics and probiotics in the development and progression of PDAC

The most recent data show that effectiveness and safety of administration of oncolytic viruses in the treatment of PDAC. Oncolytic viruses created by modified engineering of the oncolytic viruses’ genome and insertion of intended transgenes including cytokines or shRNAs may inhibit development of tumours directly and some through activation of immune responses. This treatment caused increased patient survival compared to the control group. However, using oncolytic viruses alone has not been efficient in the inhibition of tumours development [185].

There is no strong evidence of the improvement of overall survival after therapies using oncolytic viruses or microbiome. Surgical treatment, adequate chemotherapy and eventually radiotherapy lengthen the lives of patients with PDAC, however only for several months.

## 6. Summary

There are some gaps in the diagnosis and treatment of PDAC (Figure 1 and Figure 2), inducing a lot of challenges and problems requiring improvement (Table 3). The early diagnosis of PDAC, which is key and necessary to extend survival, remains challenging. It requires improvements in diagnostic technology and methodologies. Unfortunately, actually there is no role for the use of any routine biomarker in the diagnosis of PDAC.

Actually, there are some main issues that require repair to improve the diagnosis and treatment of PDAC. Firstly, it is worth enhancing our understanding of this complex disease by the organization of campaigns promoting knowledge and awareness of the disease in the societies of the countries, establishment of management plans with improvement of the role of PDAC in national cancer control programmes or changes in early recognition of PDAC through national campaigns targeting patients and physicians. These actions may contribute to earlier detection of the disease, raising survival. Secondly, it is necessary to pay special attention to symptoms, such as pain not reacting to protein pump inhibitors, back pain not associated with spine diseases. Thirdly, new correct tools for earlier diagnosis are necessary to adequately identify cancer at early stage of disease advancement. The development of novel imaging technologies to detect PDAC at a very early stage and the identification of biomarkers suitable for reliable detection may be critical. New technological solutions supporting currently available radiological and endoscopic imaging, such as artificial intelligence, as well as new imaging technologies should be extensively developed to increase the diagnostic accuracy of imaging. Another important aspect is the evolution of strategies to overcome resistance to systemic therapies by development of molecular treatment approaches targeting both tumour cells and components of the surrounding microenvironment.

## 7. Conclusions

Further clinical studies are needed to improve the diagnosis and treatment of PDAC by identifying new pharmacological targets, therapeutic options and biomarkers of the disease, as well as novel diagnostic imaging. Actually, it is necessary to develop new preventive, diagnostic and therapeutic strategies that could improve the effectiveness of PDAC treatment. The identification of new diagnostic markers may prove to be key for early diagnosis of PDAC, improving patients’ prognosis and their overall survival.

## Figures and Tables

**Figure 1 cancers-15-05577-f001:**
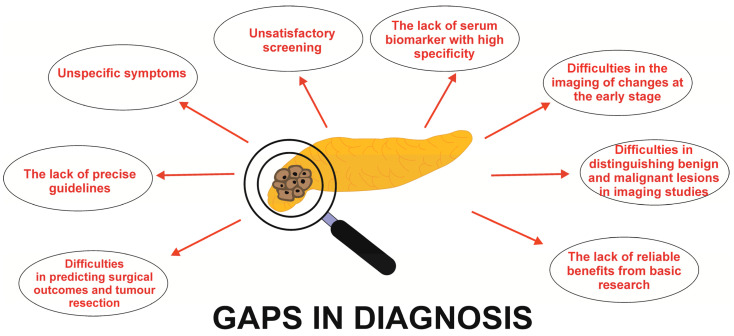
Main gaps of PDAC diagnosis.

**Figure 2 cancers-15-05577-f002:**
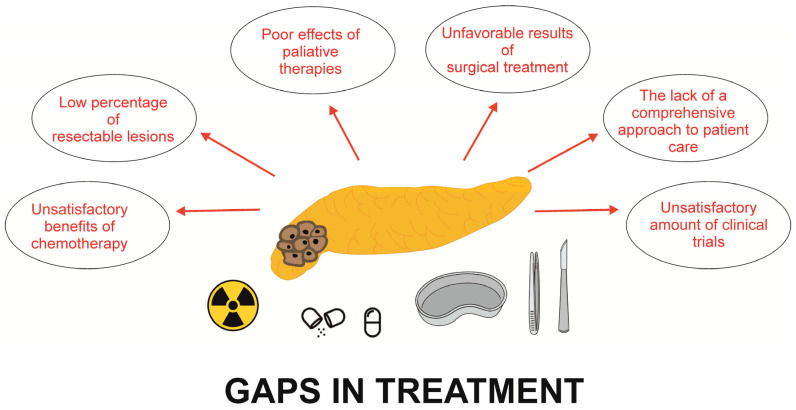
Main gaps of PDAC treatment.

**Table 1 cancers-15-05577-t001:** The risk of PC for individual genetic syndromes.

Genetic Syndrome	Risk of PC throughout Life
FAMMM	to 17%
Peutz-Jeghers syndrome	to 36%
Lynch syndrome	to 5%
Cystic fibrosis	5%
Hereditary pancreatitis	30–40%
*BRCA1*/*BRCA2* mutation	various

**Table 2 cancers-15-05577-t002:** Definition of high-risk individuals eligible for PC surveillance. It is worth emphasizing that high-risk individuals eligible for pancreatic cancer surveillance were defined for patients regardless of gene mutation status: if at least three affected relatives on the same side of the family, of whom at least one is a first-degree relative to the individual considered for surveillance or if at least two affected relatives who are first-degree relatives to each other, of whom at least one is a first-degree relative to the individual considered for surveillance or if at least two affected relatives on the same side of the family, of whom at least one is a first-degree relative to the individual considered for surveillance. ATM, ataxia telangiectasia mutated; BRCA2, breast cancer 2; CDKN2A, cyclin-dependent kinase inhibitor 2A; FAMMM, familial atypical multiple mole melanoma; LKB1/STK11, liver kinase B1/serine/threonine kinase 11; MLH1, mutL homolog 1; MSH2, mutS homolog 2; MSH6, mutS homolog 6; PALB2, partner and localizer of BRCA2.

Genetic Syndrome/Gene Mutation	Criteria of PDAC Family History
Peutz-Jeghers syndrome (*LKB1/STK11*)	Regardless of family history
FAMMM (*CDKN2A p16*)	Regardless of family history
Lynch syndrome (*MLH1/MSH2/MSH6*)	If at least one affected first-degree relative
*BRCA1*	If at least one affected first-degree relative
*BRCA2*	If at least one affected first-degree relative or at least two affected relatives of any degree
*PALB2*	If at least one affected first-degree relative
*ATM*	If at least one affected first-degree relative

**Table 3 cancers-15-05577-t003:** Table summarizing the recent challenges faced in treatment and diagnosis of PDAC.

Strategy	Challenges
Awareness	Poor awareness of the early signs and symptoms of PDAC amongst mainly older people and clinicians; Limited awareness of the burden of PDAC on the population with risk factor among clinicians; Poor awareness of the risk factor of PDAC among general population;
Detection and diagnosis	The lack of screening with proven efficacy; Limited capacities and use of comprehensive assessment to help shape of diagnosis pathways for patients with PDAC; Limited efficacy of available diagnostic methods for diagnosis of PDAC; The biology of PDAC hard to understand;
Treatment and care	Limited access to quality care in some regions of the world; Poor national development/integration of guidelines for the assessment and treatment of patients with PDAC; Limited capacities to manage patients with PDAC, especially these with multiple morbidities; Poor therapeutic response; Delays in time to approval of life-saving treatment; Insufficient development of targeted therapy;
Health system integration	Limited integration of training in oncology within core medical and nursing training; Global shortage of specialists to support a multidisciplinary treatment approach; Limited financial and other protection measures for patients with PDAC; Limited availability of oncology clinics; Variable integration of services within national health system and cancer planning; Differences in healthcare and financial outlays in various countries;
Research	Insufficient amount of clinical trials in the context of new therapeutic options with high efficacy in PDAC; Poor integration of patients with PDAC into clinical trials, which reduces their relevance; Limited translational research into the needs of patients with PDAC; Limited data on the investment-case;

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
