# Peer review of "Gaps and Opportunities in the Diagnosis and Treatment of Pancreatic Cancer"

_cancers, 2023, doi:10.3390/cancers15235577_

Round 1

Reviewer 1 Report

Comments and Suggestions for Authors

Interesting review. My comments:

1) The authors should add some comment on the potential palliative therapeutic impact of LAMS and the gaps concerning this aspect

2) The authors should comment on the gap in the diagnostic algoritm of PDAC, particularly concerning the availability of newer end-cutting FNB needles that are highly performing in this setting (cite the recent SRMA: PMID: 31031330)

3) The authors should comment on the improvements obtained with EUS in this field and the gaps concerning the optimization of tissue sampling (cite the recent papers: PMID: 33481633)

4) Some tables would be helpful to the reader

Author Response

Thank you for comments and suggestions. We agree with all comments. The responses included in attached file.

Reviewer 2 Report

Comments and Suggestions for Authors

The authors have written an interesting review article summarizing some of the challenges and new techniques for diagnosis and treatment of pancreatic cancer. However there are some areas which needs revision as mentioned below.

1. Figure 1 i.e. ‘GAPS in PC diagnosis’ could be improvised with better representation.

2. The authors could re-draw Figure 2 i.e. Main gaps of PC treatment with a better representation of the schematic diagram.

3. The authors should include a table summarizing the recent challenges faced in PC treatment and diagnosis. They could also discuss the recent interesting approach of photodynamic therapy (PDT) which is used for many cancers including pancreatic cancer. They could include a small section on it with articles such as DOI 10.1088/1748-605X/ad02d4; https://doi.org/10.1016/j.pdpdt.2020.101876.

4. The authors could also discuss the current approaches using nanomedicine or nano-theranostics which are used in pancreatic cancer to make the article more informative.

Author Response

(The authors gave the same response as above.)

Round 2

Reviewer 1 Report

Comments and Suggestions for Authors

The revised version of the manuscript is OK. Thank you!

Author Response

Thank you for comments and revision.